# Altered mRNA Expression of NFKB1 and NFKB2 Genes in Penile Lichen Sclerosus, Penile Cancer and Zoon Balanitis

**DOI:** 10.3390/jcm11247254

**Published:** 2022-12-07

**Authors:** Piotr M. Wierzbicki, Mateusz Czajkowski, Anna Kotulak-Chrząszcz, Justyna Bukowicz, Klaudia Dzieciuch, Małgorzata Sokołowska-Wojdyło, Zbigniew Kmieć, Marcin Matuszewski

**Affiliations:** 1Department of Histology, Medical University of Gdańsk, 80-211 Gdańsk, Poland; 2Department of Urology, Medical University of Gdańsk, Mariana Smoluchowskiego 17 Street, 80-214 Gdańsk, Poland; 3Hematology Laboratory, Medical University of Gdańsk, Mariana Smoluchowskiego 17 Street, 80-214 Gdańsk, Poland; 4Early Phase Clinical Research Center, Medical University of Gdańsk, Mariana Smoluchowskiego 17 Street, 80-214 Gdańsk, Poland; 5Department of Dermatology, Venereology and Allergology, Faculty of Medicine, Medical University of Gdansk, Mariana Smoluchowskiego 17 Street, 80-214 Gdansk, Poland

**Keywords:** NFKB1/2, penile cancer, penile lichen sclerosus, zoon balanitis

## Abstract

Background. The nuclear factor–κB transcription factors 1 and 2 (NFKB1 and NFKB2) are key components of the NF-κB pathway, which responds to inflammatory signals. Since the NFKB1/2 factors are activated via different inflammatory molecules, we aimed to check their expression levels in penile cancer (PC), penile dermatoses: lichen sclerosus (PLS) and zoon balanitis (ZB). Methods: Skin biopsies from altered and healthy looking foreskin were obtained from 59 (49 LS; early PLS: 13, moderate PLS: 32, severe PLS: 4; 6 PC; 4 ZB) and unchanged foreskin from 13 healthy control adult males undergoing circumcision. NFKB1/2 mRNA levels were quantified by qPCR. Results: The highest levels of NFKB1 and NFKB2 were observed in PC, ca. 22 and 3.5 times higher than in control, respectively. NFKB1 expression was correlated with PLS progression (rs = 0.667) and was ca. 20 times higher in advanced PLS than in controls and early PLS. Occurrence of micro-incontinence was associated with elevated NFKB1 levels in PLS. Conclusion: This is the first study regarding gene profiles of NFKB1/2 in PC and penile dermatoses. New drugs targeting modulation of canonical-activated NF–κB pathway should be studied and introduced to the treatment of PLS and PC apart from other treatments.

## 1. Introduction

The protein family of the nuclear factor of kappa light polypeptide gene enhancer in B-cells (NF–κB) contains five members in mammals [1]. NFKB1 (p105/p50) and NFKB2 (p100/p52) genes belong to class 1 subunits and are synthesized as longer precursor proteins that are proteolytically processed into their DNA-binding forms. In this matter, the p50 and p52 are precursors of p105 (NFKB1) and p100 (NFKB2) transcription factors, respectively. Both p105 and p100 proteins are involved in signal transduction in the NF-κB pathway [2]. In the resting cells, prior to activation, p50 and p100 proteins are connected to class 2 subunits, RelA or RelC and RelB, respectively, and are sequestered in cytosol by inhibitory IκB family proteins [1]. Activation of NF-κB via the canonical (NEMO-dependant) pathway results from cellular exposure to inflammatory cytokines such as TNFalpha and IL-1 or bacterial LPS [3]. This leads to changes in inactive NFKB1-complexes. Degradation of IκBα, followed by nuclear translocation of RelA:p50 and RelC:p50 heterodimers occurs. This results in NFKB1 activation as the transcription factor [3]. On the contrary, in non-canonical pathway (NEMO-independent), CD40 ligand or lymphotoxin β may activate IKKα. This in turn triggers proteolysis of p100 precursor to p52, while connected to RelB in the cytosol. The newly-created heterodimer RelB:p52 acts as a transcription factor in the nucleus [2]. The final result of NF–κB pathway activation leads to cell survival, further inflammation, immune response and anti-apoptosis [4]. In malignancy processes, it was noted that the activation of either NFKB1 or NFKB2 is associated with different phenotypes of cancer, namely in diffuse large B-cell lymphoma (DLBCL) [5]. Moreover, it was noted that other complexes of NFKB TFs may be formed, which are important in carcinogenesis and inflammation: namely, p50:p50 or p52:p52 homodimers. The P50:p50 complex is responsible for anti-inflammatory processes [3], while p52:p52 is involved in n cell proliferation, migration and inflammation [6].

Penile cancer (PC) is rare disease: American Cancer Society estimates for penile cancer in the United States for 2022:2070 new cases and 470 deaths from PC [7]. Squamous cell carcinoma is the most common (ca. 80%) of all PC histological subtypes [8]. Several risk factors for penile cancer were identified, such as phimosis, smoking history and high-risk human papillomavirus (HPV) types infection [8]. Penile lichen sclerosus (PLS) is a chronic and fibrotic dermatosis. Sclerosis seems to be the main factor which provides to penile LS complications such as phimosis, paraphimosis, painful erections, and urethral strictures. Risk factors associated with penile PLS development are lack of circumcision, all kinds of genital skin injury, elevated BMI, diabetes mellitus and postmicturition micro-incontinence (MI) [9]. Zoon balanitis (ZB) is rare but probably mostly underdiagnosed disease [10]. ZB is an idiopathic, chronic and benign inflammatory mucositis of the genitalia that clinically presents as solitary, shiny, well-defined erythromatous plaque of the glans [11]. Poor hygiene, repeated local infection and MI are based on ZB etiology [11,12].

Our recent study focused on the expression profile of pro-inflammatory cytokines in PLS [9]. Since we observed overexpression of pro-inflammatory cytokines in this disease, the next step was to verify the possible activation of cellular inflammatory pathways. We also decided to include rare penile dermatosis–ZB as well as malignancy–penile cancer. Therefore, the main goal of the present study was to check the basic parameter of NF–κB signaling at the mRNA expression levels of the main downstream activators, NFKB1 and NFKB2, in penile diseases. The obtained data may support the selection of new drugs which moderate pro-inflammatory cellular pathways.

## 2. Materials and Methods

### 2.1. Patients and Sample Collection

Seventy-two (72) adult male patients were enrolled in the study. The project was approved by the independent Bioethics Committee (decision No. NKBBN/369/2017) and written consent to participate in the study was obtained from all patients. Thirteen healthy males underwent surgery due to a short frenulum and were included in the healthy control group. Forty-nine (49) males suffered from PLS, six had PC and four patients suffered from ZB and underwent circumcision (or surgery of a cancer lesion) at the tertiary referral Department of Urology, Medical University of Gdańsk, Poland, between January 2017 and December 2019. To determine the presence of microincontinence (MI), the patients were asked about voiding patterns before phimosis formation, particularly regarding the leaking of tiny drops of urine from the urethral meatus after urination. The criterion for inclusion in the current project was the confirmation of the disease in anamnesis and histopathological examination (for LS, ZB, PC). The exclusion criteria were chemotherapy or other cancer therapy for patients with PC and detection of cancer cells in biopsies of patients from other groups, sexually transmitted diseases (STD: HIV/AIDS, gonorrhea, HPV, syphilis). For the control group, the exclusion criteria were the presence of dermatosis, neoplastic disease or STD. Apart from demographic data, the C-reactive protein (CRP) level was measured in the venous blood of each enrolled patient. The summary data are presented in Table 1. Adjacent biopsies were tested for the presence of HPV. Material collection and further processing were performed as described in our previous study [9]. During surgery, skin biopsies from foreskin were obtained. Specimens (ca. 3 × 3 × 3 mm) of healthy (control group) or healthy looking foreskin (PLS, PC and ZB groups) were collected. Lesion fragments of the similar sizes were also obtained from patients with PLS, PC and ZB. Specimens were immediately placed in five volumes of RNA-Later (Ambion - Thermo Fisher Scientific, Warsaw, Poland) in sterile vials, stored in a fridge for 6–24 h followed by storage at −80 °C until RNA processing. Unchanged skin fragments, which were directly adjacent to analyzed foreskin samples, were placed in buffered (pH = 7.4) 4% formaldehyde (POCh, Gliwice, Poland) fixative and stored in a fridge. Those biopsies were further processed for histopathological assessment.

### 2.2. RNA Extraction and Quantification of NFKB1 and NFKB2 Genes

Due to the high resistance of tissues to the homogenization required for efficient RNA isolation, we modified the RNA extraction [9] method of Chomczynski and Sacchi [13]. Briefly, tissues were drained from RNA-Later liquid, cut with scissors into small fragments and placed in a 1.5 mL Eppendorf tube with 800 μL Fenozol (AA Biotechnology, Gdynia, Poland). The tubes were incubated in a TS-100C (BioSan, Riga, Latvia) thermoblock at 50 °C for 45 min. After adding 200 μL of chloroform (POCh), samples were gently mixed and incubated at room temperature (RT) for 5 min, followed by centrifugation at 12,000 rpm for 15 min at 4 °C. The next steps of RNA extraction were carried out with the manufacturer’s protocol. The final elution volume was 100 μL of RNAse-free water. After RNA quantity and purity assessment (Epoch 800 plate reader), RNA was stored at −80 °C for further analysis. cDNA synthesis was performed as previously described [14]. Total RNA (2 μg) was reverse transcribed with RevertAid Reverse Transcriptase (Fermentas; Thermo Fischer Scientific, Warsaw, Poland). An amount of 1 μL of four times diluted cDNA and 0.2 µM of each primer was used in 10 μL total volume of qPCR reaction with AmplifyMe SybrGreen reagents (Blirt, Gdańsk, Poland). Primers sequences were created with the use of NCBI Primer-Blast on-line software and purchased in Sigma-Merck, Poznan, Poland: NFKB1 (NM_001165412.2): 5′-CATATTTGGGAAGGCCTGAACA; 5′-CCCACATAGTTGCAGATTTTGAC, NFKB2 (NM_001077494.3): 5′-GAGAACGGAGACACACCACTG; 5′-GGTGGATGACATAGACTATCTGC, glucuronidase beta (GUSB) (NM_000181.4): 5′-ATGCAGGTGATGGAAGAAGTGGTG; 5′-AGAGTTGCTCACAAAGGTCACAGG [9,14,15]. All reactions were run in duplicate in a StepOnePlus apparatus with appropriate software (ver. 2.3, Applied Biosystems). GUSB gene expression was used for the normalization of qPCR results [16] with Livak and Schmittgen’s 2ΔΔCq method [17]. 

### 2.3. Statistical Analysis

Statistical analysis was performed using the GraphPad Prism ver. 6.07 (Prism Software, Irvine, CA, USA) software. The following statistical tests were used: 2 × 2 Fisher’s exact, Shapiro–Wilk normality; non-parametric Mann–Whitney U, Wilcoxon signed-rank and Spearman’s correlation tests followed by Bonferroni test. A two-sided *p* < 0.05 was considered to indicate a statistically significant difference with a 95% confidence interval in all analyses.

## 3. Results

### 3.1. Patient Characteristics

Based on histopathological examination, PLS patients were divided into three stages: early PLS (n = 13; 18%), moderate PLS (n = 32; 44%) and severe PLS (n = 4; 6%). The severity division of PLS [18,19] was based on the following characteristics: early PLS was characterized by hyperkeratosis of the epidermis, loss of dermal papillae, basal cell degeneration, dense inflammatory infiltration that contains lymphocytes, macrophages and monocytes which are located directly beneath the basal cell layer and around blood vessels. In moderate PLS, massive lymphocytic presence is also observed when the epidermis with acanthosis and hyperkeratosis. It is separated by a narrow zone of hyalinization. On the contrary to early and moderate PLS, the severe stage is characterized by a thin epidermis with hyperkeratosis, thick dermal hyalinization and sparse lymphocytic infiltrate in lower dermal layers [9].

There was no difference in clinical characteristics of patients with different PLS stages (data shown in [9]). All penile cancer samples were histologically classified as usual squamous cell carcinoma, while the clinical advancement was stage I (pT1aN0Mx). Foreskin samples from healthy males and healthy looking foreskin samples from PLS, PC and ZB were histologically confirmed as morphologically unchanged and therefore treated as either control or margin. Human Papillomavirus (HPV) was not found in any sample. 

### 3.2. Expression Data of NFKB1 and NFKB2 in All Studied Groups

The highest level of NFKB1 expression was observed in tumor PC samples (ca. 22 and four times higher than in controls and normal foreskin samples, Figure 1A). Ca. 7 and 3.5 times higher expression was also found in PLS samples than in controls and normal foreskin from PLS patients, respectively. 

Just as in NFKB1 expression, NFKB2 mRNA was found as the highest levels in PC, but the observed upregulation between normal and cancer samples was not as high (ca. 3.5 times higher than in control and normal foreskin, Figure 1B). Despite higher levels of NFKB1 and NFKB2 mRNAs in PLS, there were no differences in comparison to control or normal foreskin. In the ZB samples the expression of both NFKB1 and NFKB2 mRNA levels did not differ from that in controls (Figure 1A,B).

Moreover, we found that there were no relationships between the expression of NFKB1 and NFKB2 genes between tissue samples group divided by type of disease (data not shown).

### 3.3. Gene Expression Levels and Clinical Data 

When NFKB1 and NFKB2 mRNA levels are compared in relation to age, BMI or CRP serum concentrations, no statistically significant relationships have been observed (data not shown, available if requested). However, when PLS tissue samples were analyzed in different stages, a medium/strong positive correlation (rs = 0.667, *p* < 0.0001, Spearman test) between NFKB1 mRNA levels and the severity of PLS was revealed (Figure 2A). We found that the NFKB1 gene was overexpressed approximately eight and twenty-two times higher in moderate and severe PLS than in controls (Figure 2B). In contrast to NFKB1 expression, no relationships in NFKB2 expression between groups could be established (Figure 2C,D).

### 3.4. NFKB Expression Concerning MI Status 

Most (44 out of 49) PLS patients suffered from postmicturition MI (*p* < 0.0001 between MI occurrence in PLS in comparison to patients with other diseases, respectively, 2 × 2 Fisher’s exact test). Since the occurrence of MI between PC and ZB patients was statistically insignificant, we focused on a possible relationship between MI and expression of NFKB1 and NKFB2. We found that the levels of NFKB1 mRNA were almost eight times higher in PLS patients who suffered from MI, while patients without MI had almost unchanged NFKB1 expression (Figure 3A). For NFKB2 expression, a similar pattern to that of NFKB1 was observed, however, the difference was not statistically significant (Figure 3B).

## 4. Discussion

The nuclear factor kappa B belongs to one the most important pathways that maintains and coordinates the inflammatory response of the cell, followed by cellular differentiation, proliferation and survival. Members of the NF–κB family include RelA, RelB, cRel, NFKB1 (p50) and NFKB2 (p52) monomeric proteins that form homodimeric or heterodimeric complexes with the affinity to bind and activate genomic DNA [20]. In resting cells, NF–κB-related transcription factors (TFs) are located predominantly in the cytoplasm and associate with members of the inhibitory IκB family, including IKKBα, IKKBβ, IκBδ and IKKBε [1]. Canonical activation (NEMO-dependent) of the NF-κB signaling pathway is triggered by variety of inflammatory cytokines (TNFα, IL-1, IL-6, IL-12), chemokines (e.g., CXCL1, CXCL2 and RANTES) [3] as well as pathogen-associated molecular patterns (DNA of pathogens and/or proteins, PAMS) [21,22]. This leads to ubiquitination and proteasomal degradation of IKKBα and IKKBβ proteins, thus releasing either c-Rel:p50(NFKB1) or p65:p50(NFKB1) from cytosol complexes with IKKs, respectively [3]. On the contrary, non-canonical (NEMO-independent) activation of NF–κB is driven only by TNF receptors (BAFFR, CD40, LTβR, RANK, TNFR2, Fn14) [20]. Through activation of NIK/IKK1 complex, degradation of IκBδ occurs, which results in the release of the p100 protein and its proteolysis into p52 protein, the active TF. Furthermore, p52 heterodimerizes with RelB. The complex is then translocated into the nucleus and exhibits strong transcriptional activity [20]. Activation of the NF–κB pathway leads to the promotion of the expression of over 150 target genes, which are mostly involved in immune and stress responses [23]. Although both NFKB1 and NFKB2 proteins share similar transcriptional activity, some studies reveal differences both in canonical and non-canonical activation of NF–κB. Studies on animal models reveal that NFKB1^−/−^ mice present multiple defects in both innate and adaptive immune responses to pathogens [3]. On the other hand, NFKB2^−/−^ mice are affected by thymic medullary hypoplasia and autoimmune diseases [24]. Moreover, in humans, rare mutations in NFKB2 cause an autosomal dominant human syndrome of hypogammaglobulinemia and increased susceptibility to infections, often accompanied by organ-specific autoimmunity [25]. 

The presented study focused only on NFKB1 and NFKB2 mRNA expression levels. Therefore, other members of the NF–κB family will not be discussed. First of all, the presence and impact of homodimers p50:p50 or p52:p52 in inflammation/cancer were recorded [3,6]. P50/p50 complexes via connection with Bcl-3 [26], HDAC [26] or C/EBPα [27] proteins may lead to either blockage of κB genomic targets, chromatin condensation or anti-inflammatory gene transcription, respectively [3]. Therefore, high p50 levels may lead to the suppression of local inflammation [3]. On the contrary, based on in vitro studies on HeLa and HEK 293T cell lines, it has been found that high levels of p52:p52 homodimers, triggered by Bcl-3, are positively involved in cell proliferation, migration and inflammation. On this basis, it has been suggested that aberrant activation of NFKB2 (p52) may lead to cancer [6]. 

Results obtained in the present study showed high overexpression of NFKB1 mRNA in penile cancer. Penile cancer is a rare malignancy. We were able to find only one study regarding the NF–κB pathway in PC. Yang et al. performed a software-augmented multiplatform qPCR/DNA mutation and kinases activity analysis of tumors and adjacent normal samples obtained from 11 PC patients. Among other pathways, they observed overexpression and upregulation of kinases related to the NF–κB pathway, suggesting the importance of its role in PC development [28]. When compared to this reference, our results, which indicate NFKB1 overexpression in PC, may complement their conclusions. Regarding the evident paucity of studies on NF–κB expression in PC, we noted that analyzed PC cases were diagnosed as squamous cell carcinoma (SCC) histological subtype, while observing the different location of tumors. Pennatochiotti et al. studied samples from 79 patients with oral SCC, categorized by severity (dysplasia, less and highly invasive cases). Based on immunofluorescence (IF) and qPCR methods, they observed increased NFKB1 mRNA/protein levels in advanced stages [29]. Alongside with the presented study, Fonseca et al. observed a correlation between high NFKB1 mRNA expression with the severity of oral SCC in 20 patients. They suggested the possible activation of inflammatory pathways in this malignancy [30]. In an in vitro study, Lehman et al. observed high metastatic potential in human esophageal keratinocytes during experimental modulation of the canonical NF–κB activation [31]. The presented data partially correspond with our observations of increased NFKB1 level in penile cancer tissue. However, due to the low number of cases in our study (n = 6), coupled with just one early stage of cancer (pT1aN0Mx), drawing far-reaching conclusions of NFKB1 involvement in PC progression is precluded. Furthermore, the published studies of NFKB1 expression in SCC refer to different locations (oral cavity and esophagus) and environment (saliva or possibly GERD with acidic pH in the esophagus). In contrast, PC is associated with constant contact of the cells with an obvious irritant, i.e., urine. It should be noted that our finding of increased NFKB2 mRNA level in this malignancy is a novel observation since there are no data regarding quantification of this gene in neither PC nor SCC. Only Golozar et al. checked for the occurrence of SNPs in 410 esophageal SCC. However, they did not find any SNPs in the NFKB2 gene [32]. When compared with previously mentioned results of in vitro study (6), we may conjecture a possible impact of NFKB2 (p52) on carcinogenesis development in penile tissues.

The present observations of NFKB1 overexpression in PLS may, to some extent, refer to the results of our previous research on pro-inflammatory cytokines’ gene expression in penile lichen sclerosus (PLS) [9]. The overexpression of IL-1A, IL-6, INF-γ and IL-1B (in severe cases) in the PLS-inflamed samples was observed. Moreover, there was a significant upregulation of IL-1A and IL-6 mRNA levels in samples of patients suffering from micro-incontinence [9]. Based on the NFKB1 activation via the canonical pathway [3], we indirectly confirmed that pro-inflammatory cytokines are mainly responsible for NFKB1 overexpression in PLS. Moreover, low levels of serum CRP corroborate a lack of a general immune response to possible bacterial infection. The association of NFKB1 activation with overexpression of pro-inflammatory cytokines in PLS may be explained by the possible influence of reactive oxygen species in PLS development [33] and the inflammatory process in general [34]. Paulis and Berardesca suggested that oxidative stress during PLS may contribute to DNA damage, followed by lipid peroxidation. They also suggested that epidermal atrophy and such lesions may lead to precancerous lesions [33]. Such a pathway cannot be omitted in further research on PLS and penile dermatoses. Moreover, antioxidant agents should be tested in the aforementioned mentioned diseases [33].

We suggest that high NFKB1 levels in PLS patients with MI probably results from the irritating effect of urine. The most recent results of Koudounas et al. [35] on a human model of incontinence-associated dermatitis are in line with our recent results [9]. The authors analyzed ELISA concentrations of proinflammatory cytokines in skin biofluids after exposure of the forearm of 10 healthy persons to synthetic urine or synthetic feces. They noted upregulation of IL-1α, IL-1RA, TNF-α and IL-1α /IL-1RA ratios, as well as time-dependent increase of IL-1B and observed some differences in the inflammatory mechanisms of incontinence-associated dermatitis, depending on the moisture source. On the basis of these results [35] and our previous data [9], we suggest that urine can indirectly activate the canonical NF–κB pathway, with all subsequent consequences. Although there is no data regarding this pathway and PLS, we suggest that NKFB1 overexpression may be involved in disease progression. This follows from the positive correlation between the gene level and disease severity. On the contrary, since the occurrence of micro-incontinence in PC was neither associated with NFKB1/2 nor cytokine profiles (IL-1A, IL-6, INF-γ, IL-1RN, TGFβ and IL-1B; data not published), it seems that the influence of the irritating effect of urine does not play central role in the development of penis cancer as observed in PLS. 

Zoon balanitis is a rare penile dermatosis [11]. Therefore, only four specimens have been collected in nearly two years. Nonetheless, preliminary results showed no differences in the expression levels of NFKB1 and NFKB2. It seems that this disease may be involved with inflammatory pathway(s) apart from PLS or PC. 

## 5. Conclusions

The progression of PLS may be connected to the overexpression of the NFKB1 gene as an indirect result of the irritation effect of urine, which triggers production of pro-inflammatory cytokines. In PC, both canonical and non-canonical NF–κB axes may be activated. Therefore, we suggest that further studies on NF–κB pathway in penile cancer and dermatoses are necessary. Based on obtained results, we plan to increase the panel of analyzed cytokines to a broader group of patients.

## Figures and Tables

**Figure 1 jcm-11-07254-f001:**
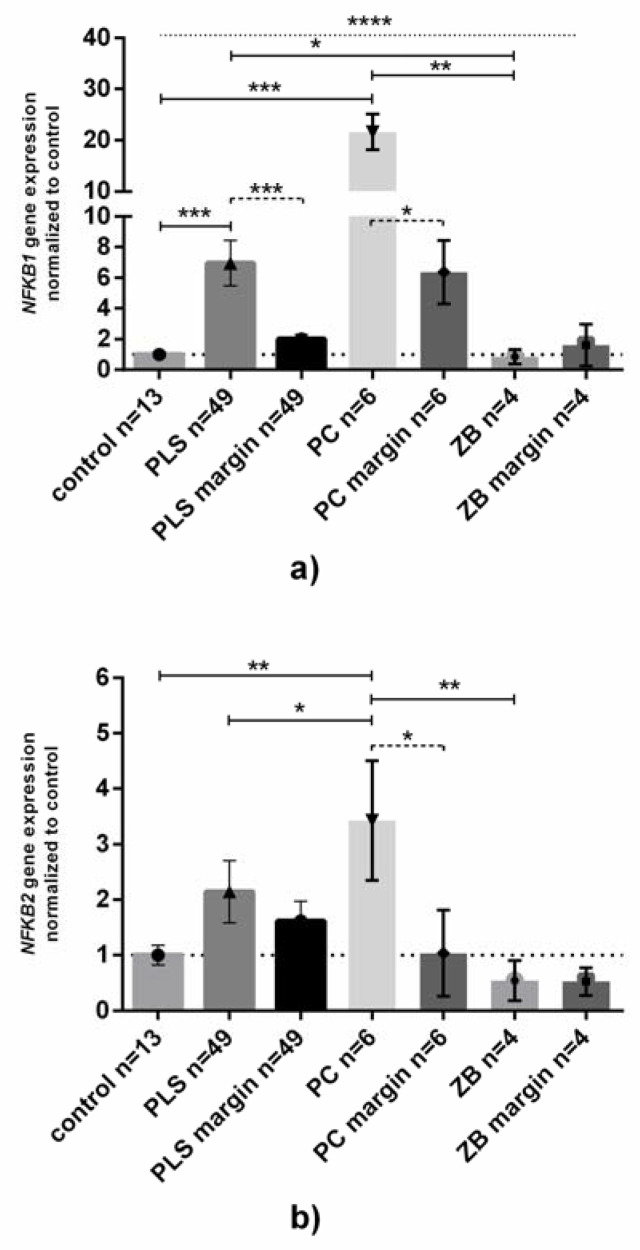
**NFKB1 and NFKB2 gene expression at the mRNA level in penile tissue.** Gene expression of (**a**) NFKB1 and (**b**) NFKB2 was assessed as described in Methods. Bars and whiskers represent the mean ± SD normalized to control foreskin samples (presented as 1). * *p* < 0.05, ** *p* < 0.01, *** *p* < 0.001, **** *p* < 0.0001 (Wilcoxon signed-rank test between inflamed-unchanged samples, dashed line; Mann–Whitney U test between each group, solid line; Kruskal–Wallis ANOVA test between all groups, dotted line). Abbreviations: PLS, penile lichen sclerosus; PC, penile cancer; ZB, zoon balanitis; margin, histologically confirmed healthy fragment of surgically removed foreskin.

**Figure 2 jcm-11-07254-f002:**
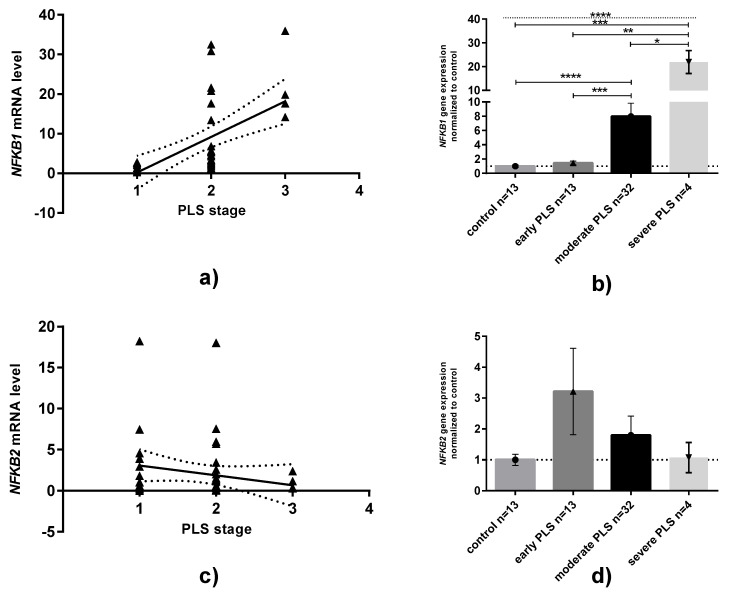
**Expression of NFKB1 and NFKB2 mRNA in penile lichen sclerosus classified in relation to the severity of disease.** (**a**,**c**): correlation plots between either NFKB1 or NFKB2 and PLS stages. Solid line: linear regression with 95% confidence interval. (**b**,**d**): NFKB1 or NFKB2 mRNA expression levels divided by disease stages: Bars and whiskers represent the mean ± SD normalized to control foreskin samples (presented as 1). * *p* < 0.05, ** *p* < 0.01, *** *p* < 0.001, **** *p* < 0.0001 (Mann–Whitney U test between each group, solid line; Kruskal–Wallis ANOVA test between all groups, dotted line). Abbreviation: PLS, penile lichen sclerosus.

**Figure 3 jcm-11-07254-f003:**
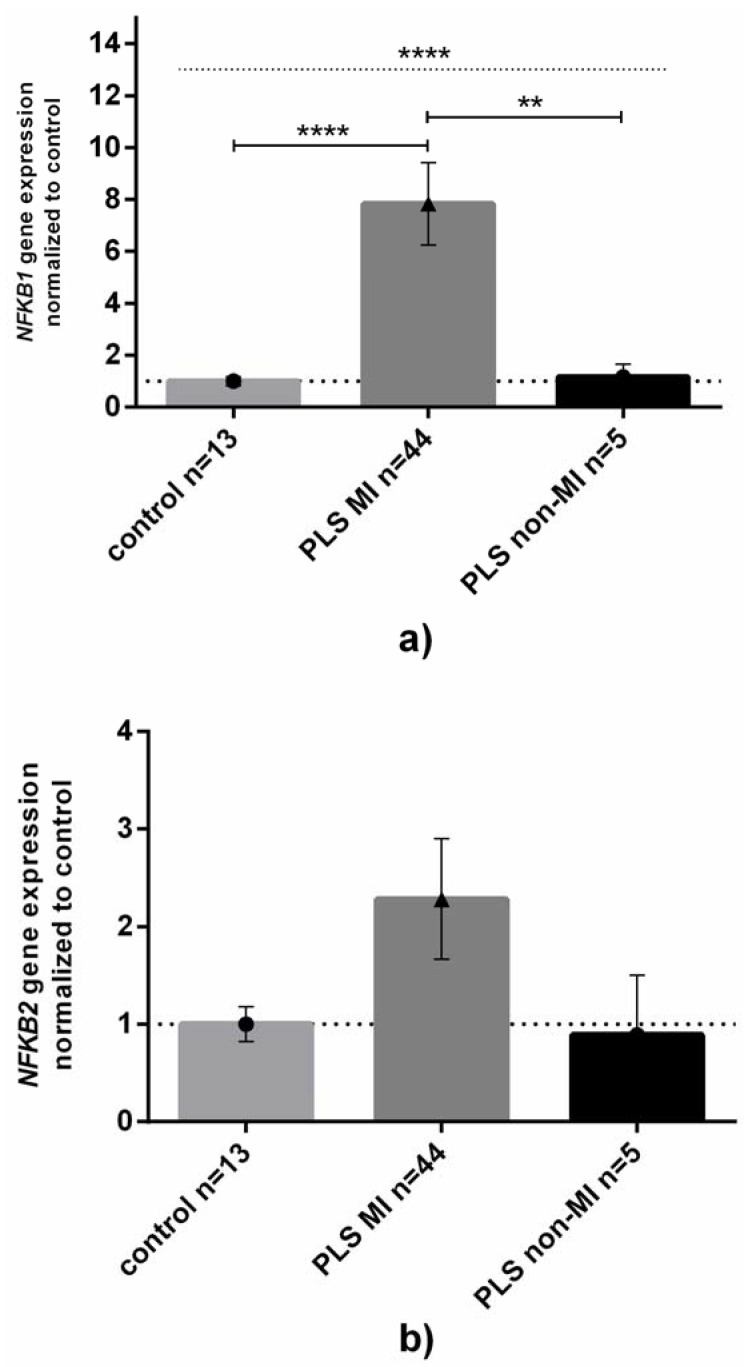
**NFKB1 and NFKB2 gene expression levels in penile lichen sclerosus related to the occurrence of micro-incontinence.** Gene expression levels of (**a**) NFKB1 or (**b**) NFKB2 in control or inflamed PLS samples were assessed as described in Methods. Bars and whiskers represent the mean ± SD normalized to control foreskin samples (dotted horizontal line at 1) grouped by each analyzed gene. Bar legends: light-grey bars represent control group (none of them had micro-incontinence), dark-grey bars represent patients with micro-incontinence (MI) while black ones represent patients without MI, respectively. ** *p* < 0.01, **** *p* < 0.0001 (Mann–Whitney U test between each group, solid line; Kruskal–Wallis ANOVA test between all groups, dotted line) between MI and lack of MI subgroups. Abbreviation: PLS, penile lichen sclerosus, MI, micro-incontinence.

**Table 1 jcm-11-07254-t001:** Clinical data of patients enrolled in the study.

Test Group	Control Group	Penile Lichen Sclerosus	Penile Cancer	Zoon Balanitis	*p* *
N	13	49	6	4	
Age [y] (min–max; mean ± SD)	19–66; 28.85 ± 14.39	20–84; 50.61 ± 17.65	37–67; 50.17 ± 11.37	21–80; 56.5 ± 25.09	0.0014
BMI [kg/m^2^] (min–max; mean ± SD)	20–43; 28.23 ± 6.89	17–39; 27.96 ± 4.95	25–31 28.00 ± 1.065	21–28; 25.75 ± 3.3	0.77
CRP [mg/L] (min–max; mean ± SD)	0.2–11.00; 1.78 ± 2.85	0.20–60.00; 4.62 ± 9.88	1.5–6.1; 3.47 ± 1.67	0.9–2.8; 1.82 ± 0.79	0.103
Micro-incontinence (YES/NO), statistics ^&^	0/13	44/5 <0.0001 ^&^	4/2 0.0039 ^&^	1/3 0.25 ^&^	<0.0001

*; *p* values assessed by a Kruskal-Wallis ANOVA test; ^&^; *p* values assessed by a 2 × 2 Fischer exact test between control group and particular groups.

## Data Availability

Normalized qPCR data available upon reasonable request.

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
