# Peer review of "Altered mRNA Expression of NFKB1 and NFKB2 Genes in Penile Lichen Sclerosus, Penile Cancer and Zoon Balanitis"

_jcm, 2022, doi:10.3390/jcm11247254_

Round 1
Reviewer 1 Report
I find the paper interesting and clear and I think it useful to have more data on the correlation between lichen and penile cancer, therefore the authors, with this paper, contribute to improving the research because they analyzed a new molecule. I recommend that other cytokines be evaluated for future studies
Author Response
Thank you for your comments. Future molecular analyzes will include a wider panel of cytokines in more patients. The relevant remark was included in the discussion.
Reviewer 2 Report
Dear authors,
Your manuscript, "Altered mRNA expression of NFKB1 and NFKB2 genes in penile lichen sclerosus, penile cancer and zoon balanitis ", shows results on the expression levels of the NFKB1 and NFKB2 genes in foreskin samples from penile cancer and dermatosis, as well as in healthy individuals. Then you compared NFKB1/2 expression levels to evaluate the severity of the lesions in these patients, as well as other clinical features. Penile cancer is a less frequent type of cancer. However, it is as heterogeneous as other more commonly diagnosed cancers. So getting a representative cohort of samples could be challenging. Despite the potential contribution of this work to the field, I would like to comment on some concerns:
Major comments
1. Regarding the heterogeneity of penile injuries. Please add inclusion/exclusion criteria for patients with penile cancer and dermatoses.
2. For patients with penile cancer, the histological subtype (usual, mixed, sarcomatoid, etc.) is not specified. Could you define it, please?
3. About your sample size. Have you evaluated their statistical power? How representative is your sample for penile lesions or tumors?
4. What type of data (dCt, -dCt, -ddCt) was showed in figures 2A,C)? It is not consistent with 2B, 2D. For example, levels of PLS stage 2 samples are lower than 8 (Figure 2A). However, in Figure 2B, the mean value of the same group is around 8. It deserves some clarification. In addition, which was the clinical/pathological criteria to classify the severity of PLS patients?
5. Although penile cancer is a rare disease, this topic has recently published studies focusing on its transcriptomic. Considering that patients in this study did not test positive for HPV, the discussion with other HPV-related tumors (p.e. head and neck) could be irrelevant. Have you evaluated HIV, HBV, or HCV in these patients? In addition, there are updated papers on transcriptomic profiles of penile lesions, mainly cancer. I suggest reviewing these studies to evaluate their inclusion in the discussion. Eventually, you can find differentially expressed miRNAs that target members of the NF-kB pathway.
6. The age of the participants has a clear impact on your sample (Table 1). Have you adjusted your p-values with this factor?
Minor comments
Typo at page 3: "StepOnePlus apparatus with appropriate d software"
Author Response
Thank you very much for your, comments. They were all very useful, we changed the manuscript as suggested.
MAJOR COMMENTS
- The following fragment was added: The criterion for inclusion in the current project was confirmation of the disease in anamnesis and histopathological examination (for LS, ZB, PC). Exclusion criteria: chemotherapy or other cancer therapy for patients with PC and detection of cancer cells in biopsies of patients from other groups; sexual transmitted diseases (STD: HIV/AIDS, Gonorrhea, HPV, Syphilis). For the control group, the exclusion criterion was dermatosis, neoplastic disease or STD.
- We checked that and all samples were examined as the usual SCC subtype (added in the text).
-
In the case of evaluating the size of the patient group, we encountered problems from a statistical point of view. Unfortunately, due to the rarity of the disease (PC, dermatoses of the penis), we included all patients (and control subjects) who agreed to participate in the study and met the inclusion and exclusion criteria in the project. In the case of penile cancer and Zoon balanitis, the numbers are close to case studies, but we performed molecular analyzes in these groups together with PLS, because we thought that separately these results would be lost in the maze of publications, or even not accepted for publication. At the same time, knowing about the small size of the groups, we used the strongest non-parametric statistical tests for the lack of normal distribution (based on Shapiro–Wilk and then; non-parametric Mann–Whitney U, Wilcoxon signed-rank, Spearman's correlation).
The study group of penile cancers is small, as are the results of worldwide studies. If you look at cancer databases that compare your results with others (e.g. cBioPortal), penile cancer is missing, probably due to the rarity of the disease. This was noted in the discussion.
- Thank you very much for this attention. Indeed, in Fig. 2A,C there was an error - these data were strictly calculated using the dCt method and were not normalized in relation to the results in the control group, therefore they do not agree with the graphs 2A, D. In the current version, the correlation graphs have been normalized as like other charts, keeping one style.
PLS severity classification is based on histological evaluation of the epidermis and dermis. The assessment is based on guidelines in two sources:
Nasca MR, Innocenzi D, Micali G. Penile cancer among patients
with genital lichen sclerosus. J Am Acad Dermatol. 1999;41:911–
914.
Billings SD, Cotton J. Inflammatory Dermatopathology. A Pathologist's
Survival Guide. New York: Springer Science + Business Media LLC;
2011.
illustrative photos of early, intermediate and severe PLS are included with the proper description in our earlier publication: DOI: 10.1007/s11255-022-03130-7
The appropriate reference for assessing the severity of the PLS roll has been added to the text.
-
We checked all the patients based on the exclusion criteria (lack of STD) and results of HBV, HCV or HPV tests. No one was sick or a carrier of the mentioned diseases.
The studies from 2015-2022 suggested above deviate from our criteria because they are based on resistance to cisplatin or are based on a mouse model. Despite the huge amount of data from these publications, we have not been able to find a starting point for the NFKB pathway we are studying. However, thank you very much for this suggestion, because it opens up new possibilities for us to study PC and dermatoses!
-
Thank you for this remark. The GraphPad v. 6.07 program we use does not include the Bonferroni test to accept/exclude the null hypothesis for several factors with p<0.05. Therefore, we calculated in another program and for 3 comparisons we have a result of p<0.0167 to recognize the validity of the hypothesis.
We carefully checked all groups regarding age and simply control group is the youngest of all (however, there is no statistical difference between them and ZB), while there is no difference between PLS and PC (p=0.95, Mann-Whitney U test).
MINOR COMMENTS
Corrected.
Reviewer 3 Report
In your study the main question addressed is relevant and
interesting. The topic of your article is very original if compared with other published articles. The paper is well-written and the text is clear and easy to read. The conclusions are consistent with the evidence and arguments presented. The conclusions address the main question posed.
There is a study in the literature where the importance of oxidative stress in lichen sclerosus has been highlighted. Since the production of NF-kB is related to oxidative stress it would be good to add this article to the References as well:
Res Rep Urol. 2019 Aug 20;11:223-232. doi: 10.2147/RRU.S205184
Antioxid Redox Signal. 2014 Mar 1;20(7):1126-67. doi: 10.1089/ars.2012.5149.
Author Response
Thank you very much for this note! That shows us an excellent goal for future studies! The appropriate section was added to the discussion with references.
Round 2
Reviewer 2 Report
Dear authors,
I thank you for responding to my previous concerns. However, I think some additional revisions are required.
1. About p-values in comparisons with healthy individuals. Table 1 shows that age is a differential factor in healthy individuals, which could affect gene expression levels. You mentioned, "When NFKB1 and NFKB2 mRNA levels have been compared with the relation to age, BMI or CRP serum concentrations, any relationships have been observed (data not shown)". Nevertheless, I think there is extremely important to show that NFKB1/2 expression is not influenced by age, especially in healthy individuals. Please add p-values (even if greater than 0.05) for NFKB1/2 and age, BMI, or CRP. Eventually, you can attach the full table showing NFKB1/2 levels after normalization and other clinical information for all anonymized patients as a supplementary table. It would help researchers who perform secondary analysis.
2. Regarding penile cancer samples and the comparison of their results. I understood your rationale for including the penile cancer group in the study. However, I think your results can be discussed in detail. The research on penile cancer is reduced due to their rarity, but there are recent studies comparing tumor and foreskin tissue from penile cancer patients without treatments or animal models. A couple of them were published in an MDPI-special issue edited (https://www.mdpi.com/journal/cancers/special_issues/ncrna_um). You could check the performance of NFKB1/2 genes in comparison with others previously described in penile cancer through a holistic vision to show the strengths and limitations of your project to propose putative penile cancer biomarkers.
Author Response
Thank you for your suggestions. We reviewed and revised our manuscript in accordance with the reviewer's concerns.
- We add p-values for NFKB1/2 and age, BMI, or CRP. We have included the data in tables 2 through 6. Moreover, we discuss these results.
- Statistical comparisons between all clinical and molecular data were performed in samples of all patients (Tables 2-6). Therefore, we observed that BMI and CRP levels increase with age (Table 2) in samples of all enrolled males. Furthermore, CRP level was middling and weakly associated with age in controls and PLS patients, respectively (Table 3 and 4). Comparison between either NFKB1 or NFKB2 mRNA levels with the relation to age, BMI or CRP serum concentrations revealed no relationships (Table 2-6).
- Regarding CRP and other demographic and clinical data of patients, we observed association between CRP, BMI and age in all, control and PLS patients. Although the result could be equivocal, it should be noted that median BMI value indicated overweight of patients in all groups. On the other hand, median CRP levels did not reach lower limit of inflammation for CRP protein. Hovewer, it is physiologically observed, that increasing age is associated with low grade chronic inflammation as assessed by elevated levels of pro-inflammatory cytokines in blood serum [37] . Although our study did not comprise such complex examination, correlation of CRP and age was observed by our team few years ago in different groups of patients [38].
2. We discuss recent research on miRNAs in penile cancer and indicate the possible direction of future studies.
- Additionally, Ayoubian et al. investigate miRNA expression in PC in relation to human papilloma virus infection (HPV), and metastatic status. They claimed that miRNAs could be the potential diagnostic and prognostic markers taking into account the division of PC for histopathological subtypes. Moreover, HPV infection was considered as an inducing factor of miRNA expression in PC [29]Similarly, Furuya et al. suggest that miRNAs (miR-432-5p; miR-487b-3p; miR-145-5p; miR-30a-5p; miR-200a-5p; miR-224-5p; miR-31-3p and miR-31-5p) and 37 genes expression takes part in penile cancer carcinogenesis, therefore would be the attractive diagnostic biomarker in future [30]. Future studies should include an analysis expression of the miRNAs and NFKB1/2 together. This may not only contribute to a better understanding of penile cancer carcinogenesis, but also find accurate diagnostic, and prognostic markers.